# Proteomic analysis of the response to cell cycle arrests in human myeloid leukemia cells

**Tony Ly, Aki Endo, Angus I Lamond***

Centre for Gene Regulation and Expression, College of Life Sciences, University of Dundee, Dundee, United Kingdom

**Abstract** Previously, we analyzed protein abundance changes across a 'minimally perturbed' cell cycle by using centrifugal elutriation to differentially enrich distinct cell cycle phases in human NB4 cells (*Ly et al., 2014*). In this study, we compare data from elutriated cells with NB4 cells arrested at comparable phases using serum starvation, hydroxyurea, or RO-3306. While elutriated and arrested cells have similar patterns of DNA content and cyclin expression, a large fraction of the proteome changes detected in arrested cells are found to reflect arrest-specific responses (i.e., starvation, DNA damage, CDK1 inhibition), rather than physiological cell cycle regulation. For example, we show most cells arrested in G2 by CDK1 inhibition express abnormally high levels of replication and origin licensing factors and are likely poised for genome re-replication. The protein data are available in the Encyclopedia of Proteome Dynamics (http://www.peptracker.com/epd/), an online, searchable resource.

## Introduction and results

Recently, we documented the chronology of protein abundance regulation across a minimally perturbed cell cycle in the human myeloid leukemia (NB4) cell line (*Ly et al., 2014*). To minimize effects on proteins caused by stress responses that do not reflect physiological regulation specific to cell cycle progression, we used centrifugal elutriation to size-separate asynchronously growing cells into populations differentially enriched in distinct phases of the cell cycle. Based on multiple criteria, including cell size, shape, and proliferation potential, the populations of elutriation-enriched cells were shown to remain within the normal range found for the starting population of non-elutriated, asynchronously growing cells. We concluded, therefore, that the variations in protein abundance measured across the elutriated cell populations predominantly reflect the physiological regulation of gene expression that occurs during the course of a 'minimally perturbed' cell cycle. We also discussed how these data may differ from many previous studies, where alternative strategies to elutriation have been used to isolate cell populations enriched for specific cell cycle phases, mostly based on methods that arrest cells at different stages of cell cycle progression. In particular, the elutriation approach identifies a smaller proportion of genes (~4%) that encode proteins whose abundance is cell cycle regulated than have been reported in previous studies.

Cell synchronization procedures, based upon metabolic and biochemical treatments that arrest cell cycle progression (including arrest-release protocols), have been more widely used to study cell cycle regulation in mammalian cells than physical methods of separation, like elutriation. Arrest methods have potential advantages, including the ease of reproducibly achieving high synchronicity. However, a major unresolved issue is the extent to which the effects of stress and metabolic perturbation, arising from the treatments used to cause cell cycle arrest, may contribute to any observed changes in protein expression unrelated to physiologically relevant cell cycle changes (*Shedden and Cooper, 2002*; *Cooper, 2005*; *Cooper et al., 2006*).

***For correspondence:**
a.i.lamond@dundee.ac.uk

**Competing interests:** The authors declare that no competing interests exist.

**Reviewing editor**: Jon Pines, The Gurdon Institute, United Kingdom

For example, hydroxyurea and serum starvation classically have been used to differentially arrest cells in distinct cell cycle phases (*Banfalvi, 2011*). Hydroxyurea depletes deoxynucleotide pools thereby slowing replication, and at high doses, arrests cells at the G1/S border (*Young and Hodas, 1964*). Serum starvation arrests cells in a state with 2N DNA content (*Zetterberg et al., 1982*). Both arrest procedures have known effects on cellular physiology. In the case of hydroxyurea, high concentrations and prolonged treatment can induce replication stress and DNA damage (*Saintigny et al., 2001*; *Alvino et al., 2007*; *Petermann et al., 2010*). Serum starvation induces reversible cell cycle exit, with cells entering a quiescent state (G0 phase) (*Pardee, 1974*; *O'Farrell, 2011*) that is characterized by changes in protein signaling, for example, pRb phosphorylation (*Chen et al., 1989*), and activation of proteins that inhibit CDK activity, for example, p21 (*Cheng et al., 2000*). Recent studies showed that the G0 state is transiently populated, even in proliferating cells cultured in serum (*Spencer et al., 2013*; *Naetar et al., 2014*). This occurs in an apparently stochastic, but p21-dependent, manner (*Spencer et al., 2013*). Additionally, microarray analyses of serum-starved mouse 3T3 cells show that up to ~5% of the probe set changes by ≥1.5-fold (*Oki et al., 2014*). However, it has not been demonstrated how hydroxyurea and serum starvation treatments affected protein levels globally and/or reflect bona fide cell cycle regulation in proliferating cells, as opposed to effects due to the arrest procedure.

Here, we have addressed experimentally how three arrest procedures affect protein expression by analyzing, at a proteome-wide level, variation in protein abundance between cells arrested in different stages of interphase using metabolic/biochemical perturbations (*Figure 1A*). To facilitate the comparison between these arrested cell populations and our previous analysis of the 'minimally perturbed' cell cycle using elutriated cells, all experiments have been performed on the same NB4 cell line (*Lanotte et al., 1991*), grown, and analysed using the same equipment and culture conditions. Serum starvation for 48 hr was used to arrest cells in G0/G1 phase. To arrest cells in S phase, cells were treated with hydroxyurea for 18 hr. To arrest cells in G2 phase, cells were treated with RO-3306 for 18 hr, which specifically inhibits CDK1 activity and arrests cells at the G2/M border (*Vassilev et al., 2006*). Each experiment was performed in biological triplicate.

Flow cytometry analysis of DNA content and immunoblot analysis of established cell cycle markers confirmed that the respective serum starvation (SS), hydroxyurea (HU), and RO-3306 treatments each arrested NB4 cells at the expected cell cycle phase (*Figure 1B,C*). Thus, the DNA content profiles show that SS increases the frequency of 2N DNA content cells (G0/G1 arrest), HU increases the frequency of cells between 2N and 4N DNA content (S arrest), and RO-3306 increases the frequency of 4N DNA content cells (G2 arrest) (*Figure 1B*). Interestingly, NB4 cells appear hypersensitive to hydroxyurea and undergo apoptosis at concentrations typically used to arrest other cell lines (i.e., 1–5 mM vs 80 μM in this study). RO-3306, which can induce genome re-replication (*Vassilev et al., 2006*; *Ma et al., 2009*), induces re-replication in ~3.3% of treated NB4 cells (cells with >4N DNA content in *Figure 1B*). As shown in *Figure 1C*, the abundance profiles of cyclin E and cyclin A, which both peak in S-arrested cells, and cyclin B1, which peaks in G2-arrested cells, are consistent with the reported cell cycle regulation for these proteins (*Pines and Hunter, 1989*; *Pines, 1999*).

In summary, the DNA content and immunoblot analyses indicate that the arrested NB4 cell populations show high enrichment for the targeted cell cycle phases. Moreover, the enrichment profiles obtained by arrest are similar to those obtained by centrifugal elutriation (cf. [*Ly et al., 2014*] *Figure 2*). Therefore, any differences observed in protein abundance between cell cycle phases in the respective arrested and elutriated samples will not be due primarily to differences in enrichment efficiency.

Lysates prepared from arrested cells were processed for quantitative, label-free proteomics essentially as described in *Ly et al. (2014)*, with differences only in the digestion protocol and peptide chromatography procedure (see 'Materials and methods' for details). In total, 46,783 peptides were identified, corresponding to 4,339 proteins identified with two or more peptides per protein (submitted to the ProteomeXchange Consortium via the PRIDE partner repository, accession PXD001610). Comparison of the label-free quantitative (LFQ) intensities (*Cox 2014*) between all three biological replicates, (*Figure 1D*), reveals a high positive correlation (>0.95) in all cases, indicating that the intensities measured are reproducibly quantitated. The data were filtered further to include only those proteins that were detected in all three replicates within a treatment group. This produces a high quality data set comprising 3,068 proteins that were used to evaluate changes in protein abundance in the respective arrested cell populations (*Supplementary file 1*).

Protein abundance measurements for each of the treatments are analysed as discrete points along a continuous axis (i.e., the cell cycle). As in the elutriated cell study (*Ly et al., 2014*), a protein was

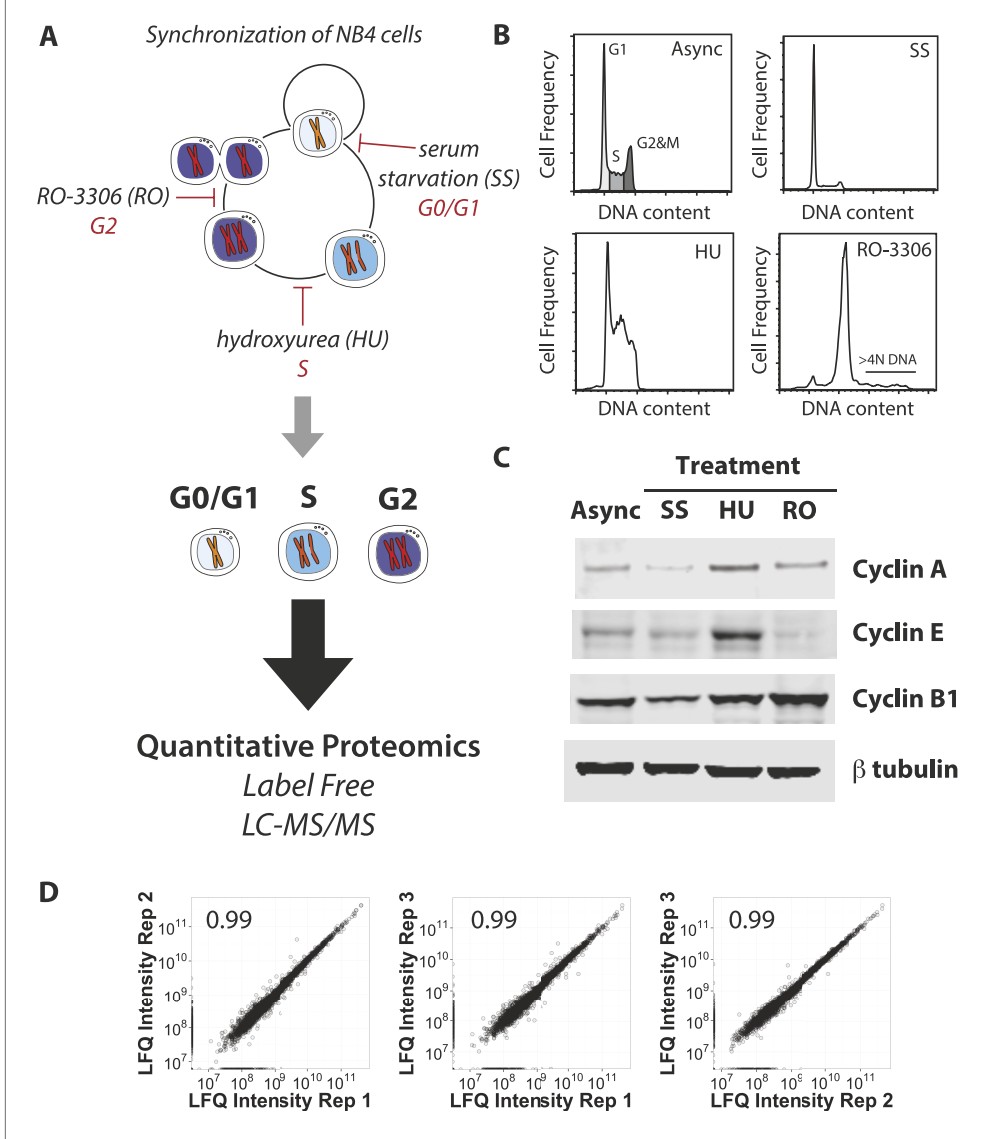

**Figure 1**. Experimental workflow and positive controls. (**A**) NB4 cells were differentially treated using serum starvation (SS), hydroxyurea (HU), and RO-3306 to arrest cells in G0/G1, S, and G2 phases of the cell cycle, respectively. Cells were then processed for label-free quantitative MS-based proteomics in a similar manner to the previous analysis of elutriated cells (*Ly et al., 2014*). (**B**) Asynchronous and arrested cells were stained with a DNA-binding dye and analyzed by flow cytometry. DNA content histograms are shown. Cells with >4N DNA content (~3.3%) are highlighted in the RO-3306 data. (**C**) Immunoblot analysis of the arrested cells using antibodies recognizing cell cycle phase-specific markers (cyclin A, cyclin E, and cyclin B1) and beta tubulin as a loading control. (**D**) Pairwise comparisons of MS-based protein abundances (LFQ intensities) independently processed and measured from three asynchronous NB4 cultures. Pearson correlation coefficients are reported in the top left corner of each scatter plot.

classified as regulated by cell cycle arrest when the protein abundance change was ≥twofold between any two conditions. The statistical robustness of the measured fold changes in the arrest data set was determined using a one-way ANOVA and a p-value cutoff of 0.05 was implemented. Thus, based on multiple criteria, including multiple supportive sequence-unique peptides, biological reproducibility, and effect size, 484 proteins were deemed cell cycle arrest regulated (*Supplementary file 2*).

The abundances of these 484 arrest regulated proteins were scaled to have the same mean and standard deviation, and the patterns grouped by hierarchal clustering and illustrated using a heatmap

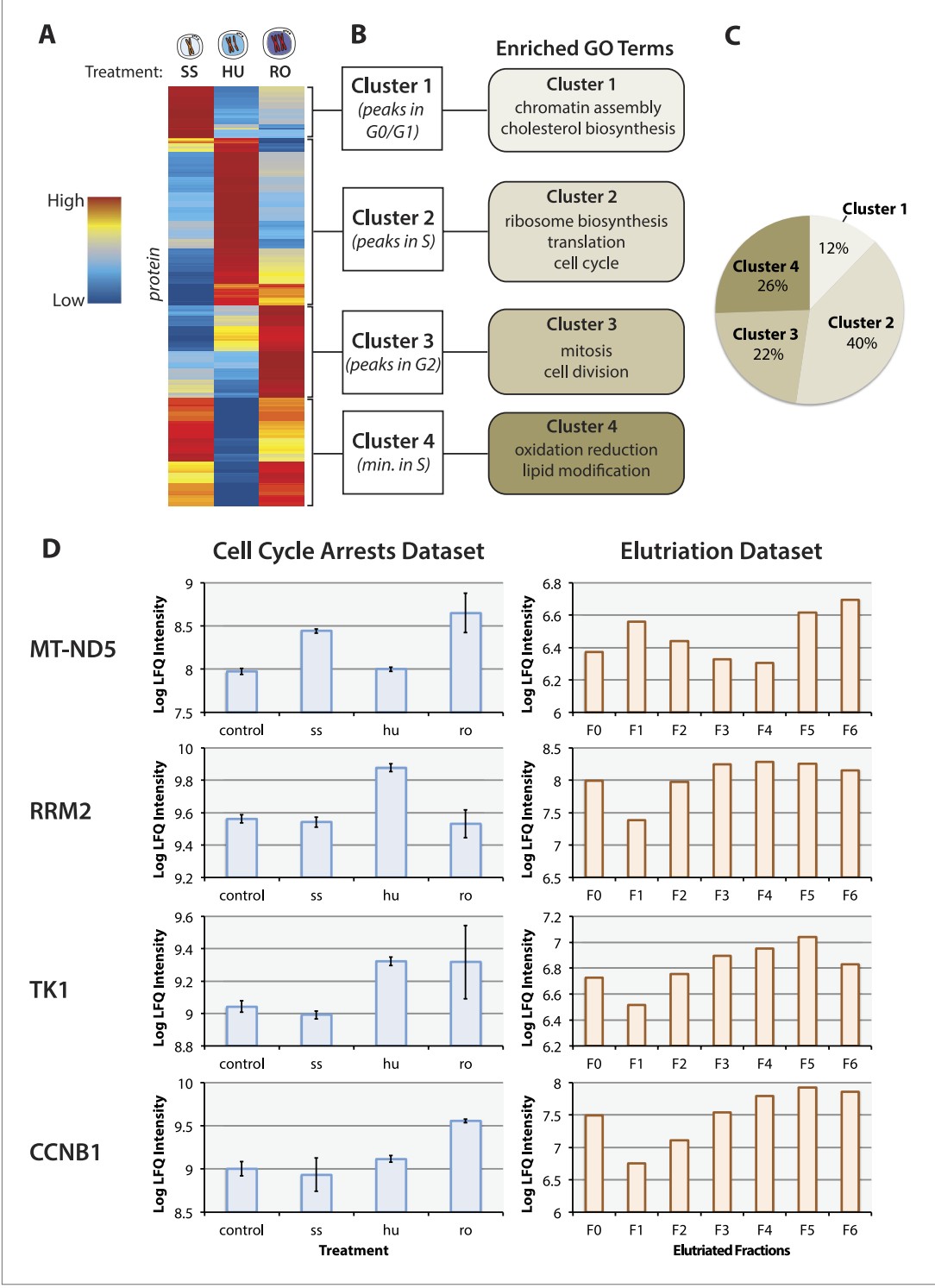

**Figure 2**. The proteomic response to cell cycle arrests. (**A**) The final proteomic dataset after quality control filtering consisted of 3,068 proteins identified with two or more peptides per protein and quantitated in all three replicates within a treatment group. 484 proteins vary in abundance between asynchronous arrested cells using cutoffs based on effect size (≥twofold change between any two conditions) and statistical robustness (p < 0.05, ANOVA). The scaled and clustered abundances of these 484 arrest regulated proteins are illustrated as a heatmap. Each protein is represented by a horizontal line, and the colour (red: high, blue: low) represents the scaled abundance in three treatments. (**B**) We identify four clusters based on peak expression, which are differentially enriched in gene

*Figure 2. Continued on next page*

*Figure 2. Continued*

ontology (GO) biological functions. (**C**) The proportions of arrest regulated proteins in each cluster. (**D**) Comparison of the protein abundance changes measured in the cell cycle arrest data set vs the elutriation data set (*Ly et al., 2014*) for selected proteins (MT–ND5, RRM2, TK1, and CCNB1). Error bars indicate the standard error of the mean log-transformed abundances.

(*Figure 2A*). This identified four clusters (*Figure 2B,C*), each showing peak protein abundance at different cell cycle phases, that is, proteins that peak in G0/G1 (12%), S(40%), and G2(22%) arrested cells, respectively. The fourth cluster (26%) contains proteins that show their lowest abundance in S phase. These data differ from the results obtained for the same NB4 cell line using elutriated cells in the 'minimally perturbed' system, where ~50% of the changing proteome in elutriated cells peaked at G2&M phases (Figure 5 in *Ly et al., 2014*). Analysis of gene ontology (GO) terms associated with each of the four clusters (*Figure 2B*), shows that the G0/G1-peaking cluster is enriched in chromatin assembly and cholesterol biosynthesis proteins, the S-peaking cluster is enriched in ribosome biosynthesis and translation proteins, the G2-peaking cluster is enriched in cell division proteins and the S-minimum cluster is enriched in redox and lipid modification proteins.

Of the 484 cell cycle arrest regulated proteins, only 212 correlate well (Pearson's correlation coefficient ≥0.50) with the patterns measured for proteins in the elutriated cell system (212/453, representing 47% of the union between the two data sets) (*Supplementary file 2*). The proteins that correlate well between the two datasets are highly enriched in genes annotated with functions in cellular division, for example, CDC20, Aurora Kinase B, and mitotic kinesins. Of these, only 29/212 (14%) change by more than twofold in both the cell cycle arrest and elutriated cell cycle datasets. For example, in both datasets the mitochondrial genome-encoded protein MT–ND5 peaks in G1 and G2, while RRM2, TK1, and CCNB1 (cyclin B1) are lowest in abundance in G1, and peak in S and G2&M (*Figure 2D*). While this indicates that proteins showing cell cycle regulation in abundance can be detected using either an arrest, or a minimally perturbing, elutriation strategy, it also highlights the major differences between these experimental systems. It is striking that most proteins that were detected here to change in abundance upon cell cycle arrest (53%) do not show cell cycle variation in their abundance in cells analysed by elutriation.

To explore this further, we examined the protein networks and signaling pathways that changed specifically upon each cell cycle arrest. Most of the proteins whose abundances change upon serum starvation do not show cell cycle stage variations in abundance in elutriated cells. Changes specific to the serum starvation arrest protocol include a dramatic increase in the levels of proteins involved in cellular metabolism (*Figure 3A*). Many are involved in catabolic pathways and in the generation of precursor and/or secondary metabolites. For example, the FAHD2A, GCDH, and BCKDHB genes all encode proteins important for the catabolism of amino acids.

Another major change to metabolism upon serum starvation is the increase in abundance of cholesterol biosynthesis enzymes. The cholesterol biosynthesis pathway is illustrated (*Espenshade and Hughes, 2007*) with enzymes catalyzing the conversion of acetyl-CoA to cholesterol shown as circles (*Figure 3B*). Over 60% of these enzymes (12/19) were quantitated in both control and serum starved cells, with 75% (9/12) of them increasing in abundance upon serum starvation, 42% (5/12) by more than twofold. Serum starvation of human fibroblasts has been shown to induce increased transcription of sterol metabolism genes, such as HMGCS1 (*Iyer et al., 1999*), through transcriptional activation by SREBP (sterol responsive element binding protein). Our data show that the increased transcription of sterol metabolism genes results in an increased abundance of the cognate proteins.

Serum starvation also induces significant changes in the abundance of chromatin components and chromatin remodelers, as shown in a network diagram of chromatin and chromatin-associated proteins that significantly change upon serum starvation (*Figure 3C*). Surprisingly, proteins in the nucleosome core, for example, histones H2A (HIST1H2AJ), H3.1 (HIST1H3A), and H4 (HIST1H4A), and the variant histone H2A.Z (H2AFX), are all upregulated upon serum starvation. Although histone levels are tightly controlled in general, several recent papers provide evidence that histone levels can be modulated under different environmental contexts and biochemical treatments (*Feser et al., 2010*; *Celona et al., 2011*; *Karnavas et al., 2014*). The increase in total histone H3 levels upon serum starvation was confirmed by immunoblot analysis of lysates from serum starved and asynchronous cells (*Figure 3C*, bottom). These data suggest an unanticipated effect of serum starvation in modulating total histone levels.

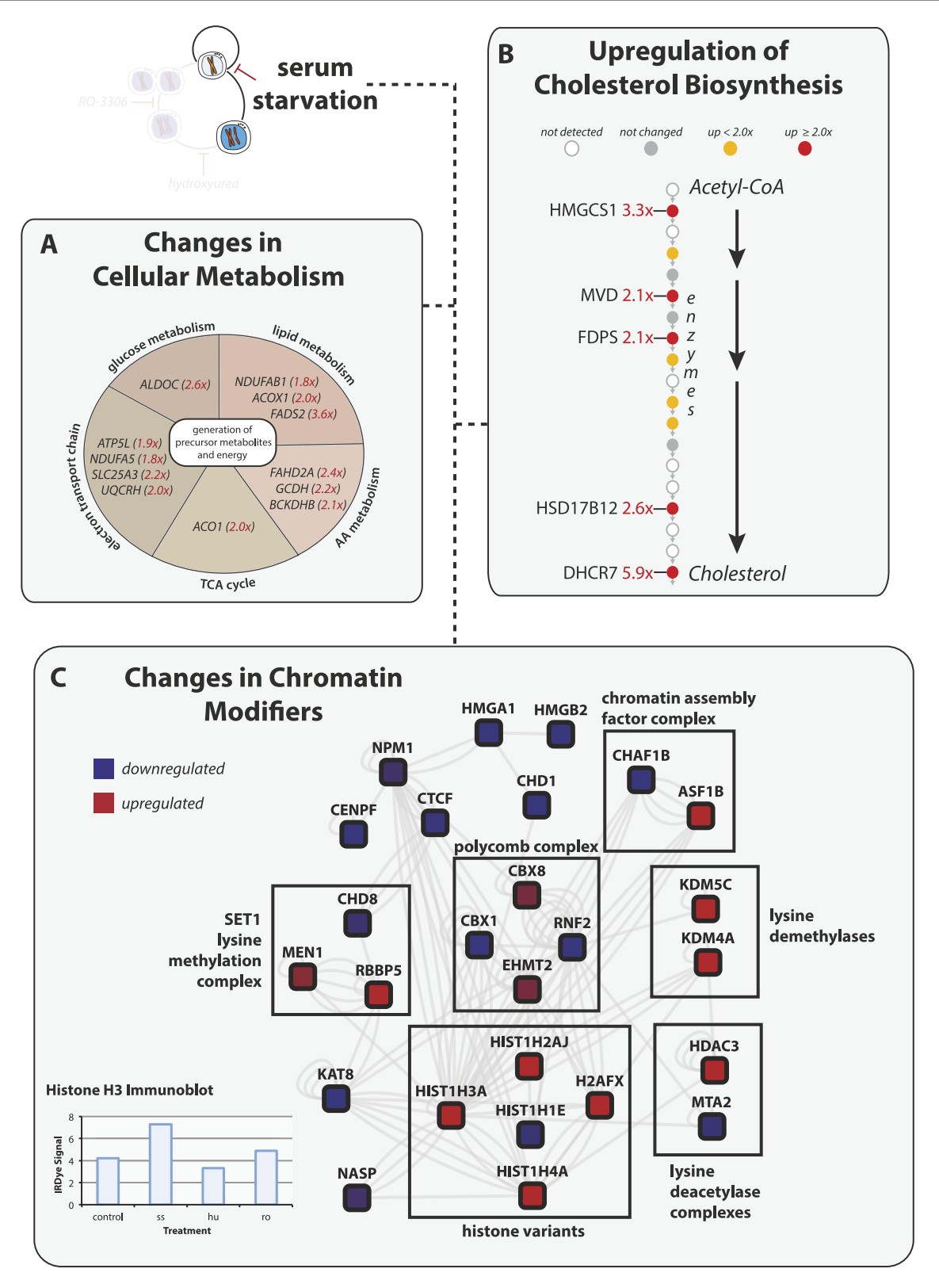

**Figure 3**. Serum starvation induces changes to cellular metabolism and chromatin remodeling proteins. (**A**) Proteins involved in generating precursor or secondary metabolites are shown grouped by metabolic pathway. Fold changes are shown in red in parentheses. (**B**) The cholesterol biosynthesis
*Figure 3. Continued on next page*

*Figure 3. Continued*

pathway is shown schematically (*Espenshade and Hughes, 2007*) with enzymes shown as circles and arrows indicating progressive steps in the pathway from acetyl-CoA to cholesterol. Fold changes are indicated by shading and are explicitly provided when greater than twofold. (**C**) Network analysis of chromatin-associated proteins that change in abundance in response to serum starvation. The colour indicates the direction of the change (red: up, blue: down), and the shading indicates the magnitude.

We also detect numerous chromatin remodelers that are changed upon serum starvation, including the histone loading CAF1 complex members (CHAF1B, ASF1B), members of the polycomb repressive complexes (CBX1, CBX8, RNF2, EHMT2) and the SET1 lysine methylation complex (CHD8, MEN1, RBBP5). Together, these protein abundance data indicate that serum starvation induces major changes affecting chromatin that are not observed during normal cell cycle progression, as judged from the elutriated cells.

Hydroxyurea arrests cells in S-phase by depleting deoxynucleotide levels and slowing replication (*Alvino et al., 2007*). Consistent with an S-phase arrest, abundances of cyclin E2, GTSE1 (G2 and S-phase expressed protein 1), and RRM2 (ribonucleotide reductase M2), all increase in abundance in S phase, relative to G1 (*Figure 4A*, highlighted in green). These changes mirror what occurs during a minimally perturbed S-phase seen in elutriated cells (*Ly et al., 2014*). However, hydroxyurea is also known to induce replication stress (*Alvino et al., 2007*) and double strand breaks (DSBs) upon prolonged treatment (*Saintigny et al., 2001*; *Petermann et al., 2010*). An unbiased pathway analysis identified the p53 signaling pathway was highly enriched amongst proteins increased in abundance after hydroxyurea arrest (*Figure 4A*). In addition to p53 itself, hydroxyurea also increases the expression of the DDB2 protein (DNA-damage binding protein 2) and the pro-apoptotic regulators BAX, APAF1, and caspase-3 (CASP3).

The protein kinases ATM, ATR, Chk1, and Chk2 are important for the repair of hydroxyurea-induced DNA damage (*Zeman and Cimprich, 2014*) (*Saintigny et al., 2001*). In mammalian cells, the role of each kinase in the response to hydroxyurea depends on dosage and cellular context (*Saintigny et al., 2001*). These four kinases are all detected in this data set with two of them, that is, CHEK2 (Chk2) and ATR, specifically increasing in abundance following hydroxyurea arrest. In contrast, in elutriated cells the abundances of CHEK2 and ATR show only minor changes, that is, less than 1.5-fold, with peak expression in S/G2-phase. These data are consistent with CHEK2/ATR induction as a response to low levels of endogenous replication stress in elutriated cells, but to a much lesser degree than induced by hydroxyurea, suggesting that DNA damage is more extensive in the arrested cells.

The small molecule, RO-3306, is a selective inhibitor of CDK1 kinase activity (*Vassilev et al., 2006*). Abrogation of phosphorylation by CDK1 arrests cells at the G2/M border and prolonged kinase inactivation results in re-replication (~3.3% of cells in this experiment) through the premature activation of the APC/C complex (*Ma et al., 2009*). Pathway enrichment analysis identified three major pathways that increase upon RO-3306 treatment. Two enriched pathways, that is, G2&M checkpoint and kinesins, are highly related to gene ontology terms enriched in the elutriated cell data set (mitosis, cell division, kinesins). Closer inspection of the enriched proteins in the G2&M checkpoint pathway, however, reveals that the majority are involved in DNA replication, such as the MCM DNA helicase (MCM6), the origin recognition complex (ORC1, ORC3, ORC5) and replication factors (RFC3, RFC5). Interestingly, ORC1 was identified in the minimally perturbed data set as a cell cycle regulated protein that peaks in G1-phase. However, ORC1 has the opposite protein abundance profile in the cell cycle arrest data set, being low in G0/G1 and high in G2 (*Supplementary files 1 and 2*).

We initially suspected that the increase of licensing and replication factors resulting from RO-3306 treatment could be due to the subset of cells that have undergone re-replication. However, given the low frequency of re-replication detected (~3.3% of cells have >4N DNA content, *Figure 1B*), we were surprised to find such a robust accumulation of licensing and replication factors. Another possible explanation is that cells with 4N DNA content express unexpectedly high amounts of origin licensing and replication factors. To test this, we measured by immunofluorescence and flow cytometry, the expression of the proteins CDT1 and geminin. Under normal proliferative conditions, the activity of CDT1, which is required for origin licensing, is inhibited in G2 phase by interactions with Geminin (*Klotz-Noack et al., 2012*). In mock-treated cells (*Figure 4C*), CDT1 protein expression is highest in G1, low in S, and accumulates in G2&M phase. This pattern is similar to what is seen in elutriated NB4

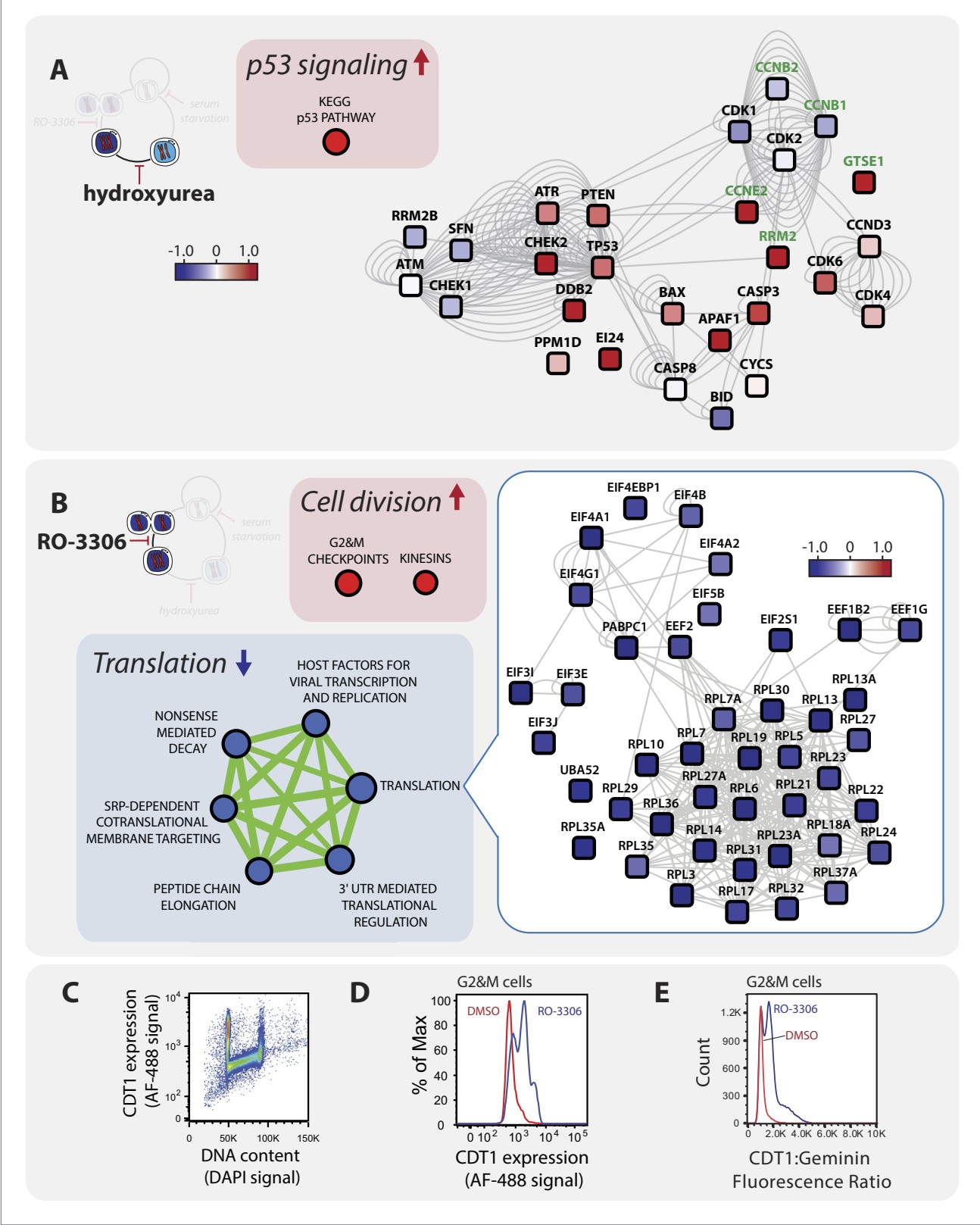

**Figure 4.** A pathway analysis of the proteomic response to (**A**) hydroxyurea and (**B**) RO-3306 treatment. Boxes containing large arrows and circles show the enriched KEGG and REACTOME pathways in each treatment. The direction of the arrows and colour indicates whether the pathway is up- (red) or

*Figure 4. Continued on next page*

*Figure 4. Continued*

down- (blue) regulated compared to asynchronous cells. The green lines and their thicknesses indicate the overlap between pathways. Proteins in individual pathways are shown as rounded squares and are connected by grey lines where protein–protein interactions have been reported. The colouring indicates the direction of the fold change and the shading represents the magnitude. (**C**) Immunofluorescence flow cytometry of asynchronous cells stained with αCDT1 antibody (AF-488 secondary conjugate, y-axis) and a DNA-binding dye (DAPI, x-axis). CDT1 protein expression is high in G1 cells, low in S-phase cells, and intermediate in G2&M cells. Comparison of CDT1 expression (**D**) and the relative ratio of CDT1:Geminin expression (**E**) measured by immunofluorescence flow cytometry in mock (red line) vs RO-3306 treated (blue line) G2&M cells gated by DNA content.

cells (Figure 8 in *Ly et al., 2014*). In contrast, upon CDK1 inhibition, G2&M cells express abnormally high levels of CDT1 protein (*Figure 4A,D*) high ratio of CDT1:geminin (*Figure 4E*) compared with mock treatment. These data suggest that CDK1 activity helps prevent premature licensing in G2. Furthermore, although ~97% of the NB4 cells are not undergoing re-replication by 18 hr post RO-3306 treatment, these data show that treated cells have accumulated high levels of origin licensing and replication factors that are not normally observed in cycling G2 cells. This indicates that the cells are likely poised for re-replication.

The abundance of many proteins involved in protein translation significantly decreases upon CDK1 inhibition (*Figure 4B*). For example, ribosomal components, eukaryotic translation initiation factors (eIF-3, eIF-4F, eIF-4B, and eIF-5B), and elongation factors (EF-1 and EF-2), all decrease upon prolonged CDK1 inhibition. These data suggest that protein translation may be impaired upon RO-3306 arrest. Comparison with the data from elutriated cells reveals that changes to translation factor protein abundance are RO-3306-specific.

## Discussion

Using label-free, MS-based proteomics, we have quantified the abundance changes of 3,068 proteins in the human NB4 myeloid leukemia cell line after arrest at different stages of the cell cycle. The arrested NB4 cells are superficially similar, in terms of their DNA content and the abundance of key cell cycle regulated proteins (cyclins), to NB4 cells enriched for the same cell cycle phases using elutriation (*Ly et al., 2014*). Comparison of these data sets from arrested and elutriated cells thus allows an evaluation of the respective proteomic responses and an estimation of the degree to which stresses arising from the specific arrest protocols may influence the observed changes in protein abundances.

While some differences between cells prepared by arrest and elutriation were to be expected, we were surprised by the scale of the differences. This study showed that the majority of the protein abundance changes observed in arrested cells are not mirrored in elutriated cells. Each arrest treatment induced major changes to the cellular proteome that are unique to that arrest method and not detected in unperturbed cells at the comparable cell cycle stage. For example, hydroxyurea increased the abundance of proteins in the p53 signaling pathway (p53, CHK2), which is consistent with previous reports showing that hydroxyurea induces a DNA damage response (*Timson, 1975*).

In contrast to hydroxyurea, which appeared to have a highly specific effect on p53 signaling, serum starvation induced changes in multiple pathways, perturbing metabolic activity and also dramatically changing the abundances of chromatin regulators and core nucleosome components. These protein abundance changes may arise, at least in part, as a response to depletion of growth factors, hormones, and lipids, which are all major components of serum. They may also reflect the shift from a proliferative to a quiescent cell state, a process that remains a subject of intense study (*Spencer et al., 2013*; *Naetar et al., 2014*; *Oki et al., 2014*). We were surprised here to find that core nucleosome components increase in abundance upon serum starvation. Increases in histone gene expression were also reported in serum-starved mouse cell lines (*Oki et al., 2014*) and in young adult Caenorhabditis *elegans* worms (*Larance et al., 2014*, submitted). These proteome changes affecting chromatin could thus represent a conserved mechanism for modulating global gene expression in response to metabolic stress caused by nutrient deprivation and merit more detailed analysis in the future. Our data set suggests that caution is warranted if the intention is to use serum starvation as a method to draw conclusions about protein abundance variations that occur in a normal, unperturbed, proliferating cell cycle.

We show that CDK1 inhibition using RO-3306 increases the abundance of key mediators of replication origin licensing, which likely contributes the DNA re-replication phenotype observed in a small percentage of treated cells (*Vassilev et al., 2006*). ORC1, a protein required for origin licensing, peaks in abundance in RO-3306 cells (4N DNA content), whereas in the elutriation data set, ORC1 peaks in

elutriated cells with 2N DNA content. We show that RO-3306 treatment increases the ratio of CDT1 to Geminin, which is normally balanced to prevent re-replication in G2 phase (*Klotz-Noack et al., 2012*). These data highlight specific pathways that are perturbed by each arrest method, likely reflecting responses to stress and/or cellular states that do not occur during a normal cell cycle, for example, G2 cells with high levels of replication factors and low CDK1 activity.

We have facilitated dissemination and community access to these data on the proteomic consequences of cell cycle arrest by depositing the data in multiple repositories targeted for different user audiences. The entire protein data set is available online via the Encyclopedia of Proteome Dynamics (http://www.peptracker.com/epd). This is a freely available, searchable resource that also includes data from multiple large-scale proteomics experiments, including measurements of protein and RNA abundances in elutriated cells across the cell cycle (*Ly et al., 2014*), protein turnover and subcellular localization (*Ahmad et al., 2012*; *Boisvert et al., 2012*; *Larance et al., 2013*), and protein complex formation (*Kirkwood et al., 2013*). For example, the EPD can be used to directly compare protein changes measured in arrested cells vs elutriated cells for a protein of interest. Additionally, we have deposited the cell cycle arrest data at intermediate stages of analysis, including the raw MS files and MaxQuant-generated output (submitted to the ProteomeXchange Consortium via the PRIDE partner repository, accession PXD001610), and supplementary tables (*Supplementary files 1 and 2*).

This study did not address proteome changes using combined arrest and release methods, such as double thymidine block and serum starvation and restoration, which are often used to synchronize cells in conjunction with cell cycle analyses. It will therefore be interesting in future to extend this study to identify also proteome changes arising from arrest and release methods and to compare these with the observed proteome changes in elutriated cells. For example, we note that serum starvation has a very acute effect on the proteome, including significant changes in proteins involved in nucleosome composition and epigenetic chromatin remodeling. It will thus be important to characterise in more detail the effects of serum starvation on chromatin structure and to investigate whether, and/or how rapidly, these effects are reversible when serum is restored. In addition to metabolic studies, we note that the MS-based proteomics approach can be used to rapidly screen cells for potential off-target effects of drug treatments, as illustrated here for RO-3306. This provides for a more detailed understanding of mechanisms regulating cell cycle progression and other processes and can also be applied in future to improve studies on cytotoxicity.

## Materials and methods

### Cell culture

The NB4 cell line was established from long-term cultures of acute myeloid leukemia blast cells grown on bone-marrow stromal fibroblasts (*Lanotte et al., 1991*). NB4 cells were obtained from the Hay laboratory (University of Dundee). Cells were cultured at 37°C in the presence of 5% $CO_2$ as a suspension in RPMI-1640 (Life Technologies, UK) supplemented with 2 mM L-glutamine, 10% vol/vol foetal bovine serum (FBS, Life Technologies), 100 units/ml penicillin, and 100 μg/ml streptomycin (100× stock, Life Technologies). Cell cultures were maintained at densities between $1 \times 10^5$ and $1 \times 10^6$ cells/ml.

### Cell cycle arrests

Cells were synchronized in G0/G1 phase by serum starvation. To starve cells of serum, cells were washed in PBS, resuspended in serum-free culture medium (with 2 mM glutamine and 10% FBS), and cultured for 48 hr in suspension before harvest. To arrest cells in S-phase, cells were treated with a final concentration of 80 μM hydroxyurea for 18 hr. To arrest cells in G2 phase, cells were treated with a CDK1 inhibitor, RO-3306 at a final concentration of 9 μM for 18 hr. All treatments were performed in triplicate for downstream MS analysis.

### Strong anion exchange (SAX) proteomics sample preparation

For protein extraction, NB4 cells were pelleted, washed twice with cold PBS, and then lysed in 0.3–1.0 ml urea lysis buffer (8 M urea, 100 mM Tris pH 7.4, Roche PhosStop, Roche, UK). Lysates were vigorously mixed for 30 min at room temperature and homogenized using a Branson Digital Sonifier (30% power, 30 s). Proteins were reduced with TCEP (25 mM in denaturing urea buffer), for 15 min at room temperature and alkylated with iodoacetamide (55 mM in denaturing urea buffer), in the dark for 45 min at room temperature. Lysates were diluted with digest buffer (100 mM Tris pH 8.0 + 1 mM $CaCl_2$) to

reach 4 M urea and then digested with 1:50 Lys-C (Wako Chemicals, Alpha Labs, UK) overnight at 37°C. The lysates were then further diluted with digest buffer to reach 0.8 M urea and digested with trypsin (1:50, Roche) for 4 hr at 37°C. Digest efficiencies were checked by SDS-PAGE analysis and Coomassie protein staining. The digests were then desalted using SepPak-C18 SPE cartridges, dried, and resuspended in 50 mM borate, pH 9.3. Peptides were separated onto a Dionex Ultimate 3000 HPLC system equipped with an AS24 strong anion exchange column, using a similar protocol to the hSAX method described previously (*Ritorto et al., 2013*). Peptides were chromatographed using a borate buffer system, namely 10 mM sodium borate, pH 9.3 (Buffer A) and 10 mM sodium borate, pH 9.3 + 0.5 M sodium chloride (Buffer B) and eluted using an exponential elution gradient into 12 × 750 µl fractions. The peptide fractions were pooled into four (F1, F2 + F3, F4 + F5 + F6, F7 + F8 + F9 + F10 + F11 + F12), desalted using SepPak-C18 SPE plates, and resuspended in 5% formic acid for LC-MS/MS analysis.

### LC-MS/MS analysis

Peptides were analyzed using a Dionex RSLCnano HPLC-coupled Q-Exactive Orbitrap mass spectrometer (Thermo Fisher Scientific, San Jose, CA). Peptides were first loaded onto a 2-cm PepMap trap column in 2% acetonitrile + 0.1% formic acid. Trapped peptides were then separated on an analytical column (75 µm × 50 cm PepMap-C18 column) using the following mobile phases: 2% acetonitrile + 0.1% formic acid (Solvent A) and 80% acetonitrile + 0.1% formic acid (Solvent B). The linear gradient began with 5% B to 35% B over 220 min with a constant flow rate of 200 nl/min. The peptide eluent flowed into a nanoelectrospray emitter at the front end of a Q-Exactive (quadrupole Orbitrap) mass spectrometer. A typical 'Top10' acquisition method was used. Briefly, the primary mass spectrometry scan (MS[1]) was performed in the Oribtrap at 70,000 resolution. Then, the top 10 most abundant m/z signals were chosen from the primary scan for collision-induced dissociation in the HCD cell and MS[2] analysis in the Orbitrap at 17,500 resolution. Precursor ion charge state screening was enabled and all unassigned charge states, as well as singly charged species, were rejected.

### Flow cytometry and immunoblotting of elutriated NB4 lysates

NB4 cells (5 × 10[5] cells, minimum) were resuspended in cold 70% ethanol and fixed at room temperature for 30 min. Fixed cells were then washed twice with PBS and resuspended in PI stain solution (50 µg/ml propidium iodide and 100 µg/ml ribonuclease A in PBS). Cells were incubated in PI stain solution for 30 min and then analyzed by flow cytometry. An asynchronous population of cells was used as a control to adjust flow cytometer settings, which then remained constant throughout analysis of the set of samples. The flow cytometry data were analyzed using FlowJo (Tree Star, Inc., Ashland, OR).

Lysates for SDS-PAGE analysis were prepared in lithium dodecylsulphate sample buffer (Life Technologies) and 25 mM TCEP. Samples were heated to 65°C for 5 min and then loaded onto a NuPage BisTris 4–12% gradient gel (Life Technologies, UK), in either MOPS or MES buffer. Proteins were electrophoresed and then wet transferred to nitrocellulose membranes at 35 V for 2 hr. Membranes were then blocked in 5% BSA in immunoblot wash buffer (TBS +0.1% Tween-20) for 1 hr at room temperature. Membranes were then probed with primary antibody overnight at 4°C, washed, and then re-probed with LiCor dye-conjugated secondary antibodies (either IRDye-688 or IRDye-800). Primary antibodies for cell cycle immunoblot analysis were obtained from Cell Signaling Technology (cyclin B1, cyclin A, cyclin E, beta tubulin, New England Labs, UK). Bands were visualized using the Odyssey CLx scanner (LiCor Biosciences, UK).

### Immunofluorescence flow cytometry

NB4 cells were fixed in 0.5% formaldehyde in PBS (pre-heated to 37°C) for 30 min at room temperature and permeabilised in ice-cold 90% methanol. One million cells were then washed in PBS, blocked with 5% BSA in wash buffer (TBS +0.1% Tween-20) for 10 min, and then stained overnight with 200 µl of primary antibody (1:200 in blocking buffer). The anti-CDT1 and anti-Geminin antibodies were obtained from Cell Signaling Technology and Abcam (UK), respectively. Cells were re-probed with Alexa Fluor dye-conjugated secondary antibodies (1:200 in blocking buffer) and then stained with 5 µg/ml DAPI in PBS prior to flow cytometry analysis on the FACS Fortessa (BD Biosciences, UK).

### Data analysis

The RAW data files produced by the mass spectrometer were analysed using the quantitative proteomics software MaxQuant, version 1.5.0.0 (*Cox and Mann, 2008*). This version of MaxQuant includes

an integrated search engine, Andromeda (*Cox et al., 2011*). The database supplied to the search engine for peptide identifications was a UniProt human protein database ('Human Reference Proteome' retrieved on 17 February 2014) combined with a commonly observed contaminants list. The initial mass tolerance was set to 7 ppm. and MS/MS mass tolerance was 20 ppm. The digestion enzyme was set to trypsin/P with up to 2 missed cleavages. Deamidation, oxidation of methionine and Gln→pyro-Glu were searched as variable modifications. Identification was set to a false discovery rate of 1%. To achieve reliable identifications, all proteins were accepted based on the criteria that the number of forward hits in the database was at least 100-fold higher than the number of reverse database hits, thus resulting in a false discovery rate of less than 1%. Protein isoforms and proteins that cannot be distinguished based on the peptides identified are grouped by MaxQuant and displayed on a single line with multiple UniProt identifiers. The label free quantitation (LFQ) algorithm in MaxQuant was used for protein quantitation. The algorithm has been previously described (*Cox 2014*). Protein quantitation was performed on unmodified peptides and peptides that have modifications that are known to occur during sample processing (pyro-Glu, deamidation). All resulting MS data were integrated and managed using PepTracker Data Manager, a laboratory information management system (LIMS) that is part of the PepTracker software platform (http://www.PepTracker.com).

Quantitative protein and peptide output from MaxQuant was analyzed using R (version 3.0.3) as implemented in the RStudio interactive development environment (version 0.98.932). In addition to the FDR thresholds implemented by default in MaxQuant, a further cleaning step was performed to improve the quality and value of the data set. This step involved removing proteins with less than 2 peptide identifications, and those labeled as either contaminants or reverse hits.

We employed two data mining strategies to identify enriched pathways and networks. The first method uses p-value and fold change thresholds to identify a set of proteins that reproducibly change in abundance. In this study, a one-way ANOVA was used to calculate the p-value. The protein set that meets the threshold criteria is then compared against the proteome to identify enriched annotations, such as gene ontology terms. The critical thresholds used were a p-value of less than 0.05 and a fold change (between any treatment) greater than or equal to two. Scaled protein abundances were then clustered into 4 clusters using the Ward algorithm (*Ward, 1963*). Enriched gene ontology annotations were identified using the DAVID web resource. The second method uses threshold-free algorithms to detect enriched gene sets. The Gene Set Enrichment Analysis (GSEA) tool (*Subramanian et al., 2005*) was used to find enriched KEGG (*Kanehisa and Goto, 2000*) and Reactome (*Croft et al., 2011*) pathways. These pathways were then visualized using Cytoscape Desktop (*Shannon et al., 2003*) and the Cytoscape plug-in, EnrichmentMap (*Isserlin et al., 2014*).

## Data sharing

The proteomic dataset is provided in multiple forms to facilitate access to a range of end-users. The mass spectrometry raw data files have been deposited to the ProteomeXchange Consortium (http://proteomecentral.proteomexchange.org) via the PRIDE partner repository with the dataset identifier PXD001610. Protein identifications and quantitations are available in a supplementary table to this manuscript *(Supplementary file 1)*. In addition, gene-by-gene visualization of the protein data set is provided in a searchable, online format via the Encyclopedia of Proteome Dynamics (EPD) (http://www.peptracker.com/epd) (*Larance et al., 2013*). This is a web-based tool that aims to visually communicate and disseminate data from large-scale multi-dimensional proteomic experiments. The EPP, which is part of the PepTracker platform, is developed using Python and the Django web framework.

## Acknowledgements

This work was supported by grants to AIL from the Wellcome Trust (Grant#:083524/Z/07/Z, 097945/B/11/Z, 073980/Z/03/Z, 08136/Z/03/Z, and 0909444/Z/09/Z), the EU EpiGeneSys network (Grant#: HEALTH-F4-2010-257082), and the BBSRC LoLa (Grant#: BB/K003801/1). We thank the UK Research Partnership Investment Fund and the Scottish Funding Council (Project H13047) for instrumentation funding. We thank our colleagues in the Lamond group for advice and discussion and in particular, Vackar Afzal for help with PepTracker Data Manager, and Robert Kent and Alejandro Brenes Murillo for help with PepTracker Datashop and the Encyclopedia of Proteome Dynamics. AIL is a Wellcome Trust Principal Research Fellow.

## Additional information

### Funding

| Funder | Grant reference number | Author |
|--------|------------------------|--------|
| Wellcome Trust | 083524/Z/07/Z, 097945/B/11/Z, 073980/Z/03/Z, 08136/Z/03/Z, and 0909444/Z/09/Z | Angus I Lamond |
| European Research Council | HEALTH-F4-2010-257082 | Angus I Lamond |
| Biotechnology and Biological Sciences Research Council | BB/K003801/1 | Angus I Lamond |

The funders had no role in study design, data collection and interpretation, or the decision to submit the work for publication.

### Author contributions

TL, Conception and design, Acquisition of data, Analysis and interpretation of data, Drafting or revising the article; AE, Analysis and interpretation of data, Drafting or revising the article; AIL, Conception and design, Analysis and interpretation of data, Drafting or revising the article

## Additional files

### Supplementary files

• Supplementary file 1. Filename: Supp_Table_1.txt. Title: Proteomics Data set Legend. This tab-delimited table summarizes the proteins identified and quantified in asynchronous NB4 cells and in serum starvation, hydroxyurea, and RO-3306 arrested cells and includes the following data for each protein identification: protein and gene identifiers, protein descriptions, the number of supporting peptides, the posterior error probabilities (PEPs), the extracted ion chromatogram (XIC) intensities, the LFQ-normalized intensities, and the p-value results from the statistical tests. P-values were calculated for pairwise comparisons between asynchronous and each of the arrest treatments ('ss.pvalue', 'hu.pvalue', 'ro.pvalue'). An ANOVA was also used to calculate a global p-value ('arrest.pvalue').

• Supplementary file 2. Filename: Supp_Table_2.txt. Title: Proteins Regulated by Cell Cycle Arrest Legend. This tab-delimited table lists the proteins that meet the fold change (≥twofold) and p-value (<0.05) criteria. Logical columns indicate whether the protein meets the cutoffs separately for each treatment versus asynchronous ('ss.changing', 'hu.changing', 'ro.changing'), and globally using the global p-value calculated from the ANOVA ('arrest.changing'). The cluster membership ('cluster') and the peak treatment ('peakfraction', 1:ss, 2:hu, 3:ro) for each protein are provided. A correlation coefficient (Pearson) is calculated between the elutriation and arrest abundance profiles ('correlation_with_ elutriation_data'). Each protein was also categorized as being either, 'Not Detected in the Elutriation Data set', 'Detected in the Elutriation Data set', 'Quantitated in the Elutriation Data set', or 'Cell Cycle Regulated in the Elutriation Data set', as indicated in the column 'status_in_elutriation_dataset'.

### Major dataset

The following dataset was generated:

| Author(s) | Year | Dataset title | Dataset ID and/or URL | Database, license, and accessibility information |
|-----------|------|---------------|------------------------|--------------------------------------------------|
| Ly T, Endo A, Lamond Angus I | 2014 | Proteomics dataset | PXD001610; http://proteomecentral. proteomexchange.org | ProteomeXchange Consortium. |

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
