## [Decision Letter]

Thank you for sending your work entitled “Proteomic Analysis of the Response to
Cell Cycle Arrests in Human Myeloid Leukemia Cells” for consideration at
*eLife*. Your Research advance has been favorably evaluated by Tony
Hunter (Senior editor) and 3 reviewers, one of whom is a member of our Board of
Reviewing Editors.

The Reviewing editor and the other reviewers discussed their comments before we reached
this decision, and the Reviewing editor has assembled the following comments to help you
prepare a revised submission.

This study builds upon the authors' previous mass spectrometry analysis of the cell
cycle in elutriated cells. In this study they have compared the changes in the proteome
in cells treated by serum starvation, hydroxyurea and the Cdk1 inhibitor RO3306. The
authors use these treatments to obtain cells in G0/G1, S and G2 phase. They find very
significant differences in proteins enriched using these treatments compared to
elutriation.

This follow up study contains some useful but perhaps not unexpected data. While the
idea of comparing the proteome in drug-arrested cells to elutriation is certainly
worthwhile, the choice of conditions used does not throw as much light on the drawbacks
to cell synchronisation techniques as the authors suggest. For example, serum-starvation
is not generally used to obtain G1 phase cells: most cell cycle researchers are aware
that G0 phase cells are in a very different state from G1 cells, as shown by the classic
studies by Anders Zetterberg. Thus, it is not particularly informative to compare serum
starved cells to G1 phase cells; it would have been more useful to add back serum and
analyse the cells after they had committed to another round of replication.

Similarly, hydroxyurea is more commonly used to induce DNA damage or replication stress.
Most mammalian synchronisation regimes use a thymidine block and release protocol
because cells recover from this more readily than from HU treatment.

The RO3306 data are more interesting because they give an insight into the role of Cdk1
in preventing re-replication.

Similar work has previously been done in yeast and using transcript profiling; this work
should at least be cited. It would have been interesting to go further and correlate the
findings between the studies. It would also be interesting to know if the authors have
performed GO analysis separately for the set of proteins that correlate well with the
elutriation experiment and the ones that do not, to try to better identify those
processes in which changes in protein abundance due to the arrest method are most
affected. In a similar vein, a comparison to an asynchronous population should be
included to determine the enriched proteins in each phase.

The one experimental concern lies with the observation of histone upregulation in
serum-starved cells. This goes against what has been reported on the very tight control
of histone levels. It is possible that this particular observation is the result of an
overcompensation of total protein levels prior to mass spec analysis. Since G0/1 cells
are likely much smaller and of lower protein content than S or G2 phase cells, it could
be that the high abundance of histones in serum starved samples only reflects a higher
number of cells. This issue should be addressed or commented upon as appropriate.

In summary, the author should re-write their paper to de-emphasise the importance of
their results for interpreting cell synchronisation techniques and put the results in
context particularly with respect to the non-equivalence of G0 and G1 phases, and the
DNA damage induced by hydroxyurea treatment.

---

## [Author Response]

We are pleased that the reviewers provided a positive assessment of our study,
recognizing the value of extending our previous analysis of cell cycle variation in
protein expression to include here measurements of protein levels in NB4 cells arrested
at different stages of interphase, which we compare with data from elutriated NB4 cells
at the same stages of interphase. We respond below to the specific comments provided by
the reviewers, explaining how we have revised the manuscript to address these points. In
addition to the points raised by the reviewers, we have also improved the revised
manuscript by including an additional figure panel (Figure 4) documenting the ratio of CDT1:geminin protein expression in cells
treated with RO‐3306. This supports an expanded discussion of our proteomic
characterization here of the effects of the CDK1 inhibitor RO‐3306 on arrested
cells, which the reviewers highlighted as one of the most novel aspects of the
study.

*This study builds upon the authors' previous mass spectrometry analysis of
the cell cycle in elutriated cells. In this study they have compared the changes in
the proteome in cells treated by serum starvation, hydroxyurea and the Cdk1 inhibitor
RO3306. The authors use these treatments to obtain cells in G0/G1, S and G2 phase.
They find very significant differences in proteins enriched using these treatments
compared to elutriation*.

*This follow up study contains some useful but perhaps not unexpected data. While
the idea of comparing the proteome in drug-arrested cells to elutriation is certainly
worthwhile, the choice of conditions used does not throw as much light on the
drawbacks to cell synchronisation techniques as the authors suggest. For example,
serum-starvation is not generally used to obtain G1 phase cells: most cell cycle
researchers are aware that G0 phase cells are in a very different state from G1
cells, as shown by the classic studies by Anders Zetterberg. Thus, it is not
particularly informative to compare serum starved cells to G1 phase cells; it would
have been more useful to add back serum and analyse the cells after they had
committed to another round of replication*.

*Similarly, hydroxyurea is more commonly used to induce DNA damage or replication
stress. Most mammalian synchronisation regimes use a thymidine block and release
protocol because cells recover from this more readily than from HU
treatment*.

We agree that serum starvation and hydroxyurea are both classic techniques for cell
synchronization, and are aware there are several methods, such as thymidine block and
release, that are now more commonly used in recent cell cycle studies. However, we feel
it is important to note that the arrest treatments we examine in this study are still
being used to synchronise cells in conjunction with cell cycle analyses, as can be seen
by performing a PubMed search for the terms ‘hydroxyurea’ and
‘serum starvation’. We also note that these arrest treatments are featured
in recent books (e.g., Methods in Molecular Biology) and review papers on
synchronization methods. We have explicitly discussed in the revised manuscript the use
of other synchronization methods not covered in this study and indicate our intention to
address also the proteomic consequences of arrest and release procedures in future
studies. However, given the previous and ongoing use of the classic arrest procedures in
the cell cycle literature, we feel that it is useful and relevant to provide this
analysis of how hydroxyurea and serum starvation affect protein expression at a global
level.

Regarding the novelty of this study, while we are aware of previous literature on the
physiological and biochemical effects of hydroxyurea and serum starvation, we believe
our present study represents the first time that the global proteome response to these
treatments has been examined quantitatively and compared with data from elutriated cells
at the same stages of interphase. We feel this allows for an unbiased and novel analysis
of the effects of these arrest treatments that can begin to unravel to what extent
changes in protein expression reflect bona fide cell cycle regulation as opposed to
effects of metabolic perturbation that do not occur during physiological progression of
cells through interphase. Reassuringly, our unbiased proteomic analysis on arrested NB4
cells is highly consistent with previous findings (e.g., effects of serum starvation on
metabolism and hydroxyurea on replication stress). We feel this validates the proteomic
methodology used here and illustrates how it can be applied in future to characterize
the effects of drugs and other treatments on cellular physiology. However, while our
data for cells arrested with either serum starvation or hydroxyurea are in general
consistent with previous studies, we note that an interesting new finding is that serum
starvation also induces changes to chromatin modifiers and chromatin components. As the
reviewers point out (below), changes in core histone levels were unexpected. This
demonstrates how our unbiased proteomic approach can reveal an unanticipated, and
previously undocumented, effect of serum starvation on cells.

*The RO3306 data are more interesting because they give an insight into the role
of Cdk1 in preventing re-replication*.

We agree that the RO‐3306 data are particularly interesting and represent a
timely component of this study. RO‐3306 is a recently developed Cdk1 inhibitor
that is now being more widely used to synchronize cells in G2. We agree with the
reviewers that our proteomic data on cells treated with RO‐3306 reveal an
important role of Cdk1 in preventing re‐replication in G2. To expand and
emphasize this point, we have included additional data in the revised manuscript that
support this conclusion. Specifically, we have performed additional quantitative
measurements of the Cdt1:geminin ratio, which has been shown to be critical for
regulating re‐replication in G2. We show that this ratio is abnormally high in
Cdk1‐inhibited G2 cells, compared with control, vehicle‐treated cells,
consistent with RO‐3306‐induced re‐licensing. We have also
increased the focus of the study towards the analysis of RO‐3306 by modifying the
Abstract and expanding the relevant section of the Discussion.

*Similar work has previously been done in yeast and using transcript profiling;
this work should at least be cited*.

We agree and have now included citations to several transcriptome profiling experiments
in yeast and in mammalian cells.

*It would have been interesting to go further and correlate the findings between
the studies*.

We agree in principle but plan to address this more comprehensively in future when we
can compare the published transcript data with our protein datasets, including our
planned future exploration also of arrest and release methods with increased proteomic
depth.

*It would also be interesting to know if the authors have performed GO analysis
separately for the set of proteins that correlate well with the elutriation
experiment and the ones that do not, to try to better identify those processes in
which changes in protein abundance due to the arrest method are most
affected*.

GO analysis was performed separately for the set of proteins that correlate well between
elutriation and the arrest datasets. The only significantly enriched functions that
emerged were ‘mitosis’, ‘cell division’, etc. Similarly, GO
analysis was performed on the proteins that do not correlate well. All these GO analysis
results are shown in Figures 3 and 4, which
illustrate the biological processes that are specifically modulated by each arrest
method and were not detected to be changing in elutriated cells.

*In a similar vein, a comparison to an asynchronous population should be included
to determine the enriched proteins in each phase*.

Proteome measurements were also made in asynchronous cells in this analysis. Fold
changes and implementation of fold‐change cut-offs were based on pairwise
comparisons between each treatment and an asynchronous population. Similarly,
p‐value calculations include measurements made in an asynchronous population
where available. The supplementary table that is provided in the revised manuscript
provides separate columns that indicate whether a protein satisfies both fold change and
p‐value cut-offs for each treatment.

*The one experimental concern lies with the observation of histone upregulation
in serum-starved cells. This goes against what has been reported on the very tight
control of histone levels*.

Although histone levels are tightly controlled in general, several recent papers, which
we now reference in the revised manuscript, provide exciting evidence that histone
levels can be modulated under different environmental contexts and biochemical
treatments (Celona et al. PLoS Biol 2011; Feser et al. Mol Cell 2010; Karnavas et al.
Frontiers Physiol 2014).

*It is possible that this particular observation is the result of an
overcompensation of total protein levels prior to mass spec analysis. Since G0/1
cells are likely much smaller and of lower protein content than S or G2 phase cells,
it could be that the high abundance of histones in serum starved samples only
reflects a higher number of cells. This issue should be addressed or commented upon
as appropriate*.

While we recognize that G0/G1 cells are smaller and therefore likely have lower protein
content than either S, or G2 phase cells, we consider overcompensation is unlikely and
further note that in any event this is not different for the arrested and elutriated
cells that we are comparing. Thus, in elutriated cells, we observed by MS analysis that
the levels of core histone proteins, as a proportion of total cellular protein content,
do not significantly change across the cell cycle. In contrast, in arrested cells, which
were processed for MS analysis in the same manner as elutriated cells, we observe a
significant, reproducible change in core histone proteins.

*In summary, the author should re-write their paper to de-emphasise the
importance of their results for interpreting cell synchronisation techniques and put
the results in context particularly with respect to the non-equivalence of G0 and G1
phases, and the DNA damage induced by hydroxyurea treatment*.

We agree that the importance of our results does not need further embellishment. We had
not intended to comment on all cell synchronization methods and have therefore
re‐focused the Abstract and text accordingly. We hope that it is now apparent
that we are only commenting here on our analysis of two classic methods of
synchronization, namely serum starvation and hydroxyurea, and on a relatively new
synchronization method using RO‐3306. Additionally, we have revised the
Introduction to include a more detailed description of the literature on the
physiological effects of hydroxyurea and serum starvation, which we hope will provide
better context for readers to interpret the results of our unbiased, system‐wide
analysis. We trust that these data will be useful to the community and seen as part of
an ongoing project to apply state of the art quantitative proteomics methods to
systematically characterize the global regulation of the proteome as cells progress
through the cell cycle.